# Comparison of impact accelerations between injury-resistant and recently injured recreational runners

**Aoife Burke**[1,2]*, **Sarah Dillon**[1,2], **Siobhán O'Connor**[1,3], **Enda F. Whyte**[1,3], **Shane Gore**[1,2], **Kieran A. Moran**[1,2,3]

**1** School of Health and Human Performance, Dublin City University, Dublin, Ireland, **2** Insight SFI Research Centre for Data Analytics, Dublin City University, Dublin, Ireland, **3** Centre for Injury Prevention and Performance, Athletic Therapy and Training, Dublin City University, Dublin, Ireland

* aoife.burke@dcu.ie

**Data Availability Statement:** The dataset used for this investigation is available and freely accessible at https://doi.org/10.34740/kaggle/dsv/3836131.

## Abstract

### Introduction/Purpose

Previous injury has consistently been shown to be one of the greatest risk factors for running-related injuries (RRIs). Runners returning to participation following injury may still demonstrate injury-related mechanics (e.g. repetitive high impact loading), potentially exposing them to further injuries. The aim of this study was to determine if the magnitude (Peak$_{accel}$) and rate of loading (Rate$_{accel}$) at the tibia and sacrum differ between runners who have never been injured, those who have acquired injury resistance (runners who have not been injured in the past 2 years) and those who have been recently injured (RRI sustained 3–12 months ago).

### Methods

Runners completed an online survey capturing details of their RRI history over the previous 2 years. Never injured runners were matched by sex, quarterly annual mileage and typical training speed to runners who had acquired injury resistance and to runners who had been recently injured. Differences in Peak$_{accel}$ and Rate$_{accel}$ of the tibia and sacrum were assessed between the three groups during a treadmill run at a set speed, with consideration for sex.

### Results

A total of 147 runners made up the three injury status groups (n: 49 per group). There was a significant main effect of injury status for Peak$_{accel}$ and Rate$_{accel}$ at the sacrum, with recently injured runners demonstrating significantly greater Rate$_{accel}$ than never injured and acquired injury resistant runners. There was also a significant main effect for sex, with females demonstrating greater tibial Peak$_{accel}$, sacrum Peak$_{accel}$ and Rate$_{accel}$ than males.

**Funding:** This study has emanated from research supported by Science Foundation Ireland (SFI) under grant number SFI/12/RC/2289_P2, cofounded by the European Regional Development Fund. The authors thank the participants for their time in partaking in this study.

**Competing interests:** The authors declare that they have no competing interests.

## Conclusion

Rate$_{accel}$ at the sacrum distinguishes recently injured runners from never injured runners and runners who may have acquired injury resistance, potentially highlighting poor impact acceleration attenuation in recently injured runners.

## Introduction

Recreational running is consistently reported as one of the most popular activities globally [1]. Running-related injuries (RRIs) are a prevalent issue however, with RRI prevalence rates of 66% reported in recreational runners [2]. Retrospective studies have made up a substantial proportion of the research exploring RRIs and their potential risk factors [3–5], likely due to the lower time and cost constraints associated with this type of research. One consistent risk factor which has been found to relate to subsequent injury has been a history of injury within the previous 12 months [6–10]. It is thought that these runners no longer exhibit the acute effects of the injury itself, but may still maintain some related factors of the injury during this time, potentially contributing to a reinjury [11]. Analysis of these runners may provide an insight into the potential mechanisms of RRI occurrence. Another running group of note are runners who have fully recovered from injury, but have not suffered any subsequent injuries (e.g. > 2 years since their most previous injury). These runners appear to have acquired an injury resistance, and may be less likely to have maintained the related factors of their previous injury [12], or perhaps have adopted a more injury resistant running technique. Finally, a third group of interest would be those runners who have never been injured. With a high life-time incidence of RRIs reported (> 90%) [13], this minority, but perhaps very insightful group, appear to have a smaller risk for injury compared to the aforementioned groups (recently injured runners and injury resistant runners). Only one study has previously compared these three groups [12], but the focus of this study was in clinical measures of strength and mobility rather than impact acceleration.

From a biomechanical perspective, repetitive forces which overload musculoskeletal structures are responsible for the breakdown of tissue and resultant injury [14]. Studies investigating the nature of these repetitive forces and their potential role in causing RRIs have frequently analysed the magnitude and rate of vertical Ground Reaction Force (vGRF). However, there is little evidence to confirm that passive (impact) or active vGRF peaks have a relationship with RRIs [5, 15–17], although there is some evidence to suggest that the rate of loading may have a relationship with specific RRIs, such as tibial stress fractures and plantar fasciitis [16–18]. One potential limiting factor of these findings is the means by which impact loading has been assessed, with force plate analysis providing a summary measure of loading on the body as a whole, failing to account for the distribution of load at specific segmental levels [19]. A solution to this is the use of wearable accelerometer sensors, which provide a low cost, light weight, localised segmental analysis and user-friendly alternative to force plates and instrumented treadmills [20–22]. Tibial accelerations in particular have been the most popular focus of segmental load analysis when exploring the relationship between impact acceleration and RRIs [3, 16, 23–25], with some evidence to suggest they are effective in discerning between injured and uninjured runners [3, 16]. However, impact accelerations at the sacrum have rarely been assessed despite the prevalence of lower back and hip injuries experienced in runners [26, 27]. In addition, the focus of impact accelerometry studies has been on the magnitude of acceleration without consideration of the rate, even though the rate of vGRF has been shown to relate

to RRIs [16–18]. As impact accelerometers have been found to be reliable measures of loading [28], particularly with reference to the magnitude of loading at the tibia, this may have influenced the choice of methods in studies investigating the relationship between impact acceleration and RRIs. More recently, the magnitude and rate of acceleration at both the tibia and sacrum have been found to be acceptable for injury-related research [29], and so the examination of both segments is warranted.

There is a dearth of research in the area of impact acceleration and RRIs in male runners. There has been trends to suggest that female runners with a history of stress fracture tend to run with greater tibial peak impact acceleration than uninjured females [3, 16]. Few studies have included males in their samples [23, 25], with the majority of studies exclusively looking at female runners [3, 16, 24]. Thus, it cannot be determined if the trends suggesting a link between peak acceleration and RRI in females are transferable to male running groups; research involving large cohorts of males is clearly required.

The aim of this study was to determine if the magnitude (Peak$_{accel}$) and rate (Rate$_{accel}$) of impact acceleration across two segments (tibia and sacrum) differs between runners who have never been injured, those who have acquired injury resistance (runners who have not been injured in the past 2 years) and those who have been recently injured (returned to running following an RRI sustained 3–12 months ago). Furthermore, given that sex has been shown to potentially be a non-modifiable risk factor for specific RRIs, a secondary aim was to determine if the difference in impact acceleration between the injury groups was different for male and female runners.

It is hypothesized that runners who have never been injured will demonstrate significantly lower impact acceleration (Peak$_{accel}$ and Rate$_{accel}$) compared to runners who have recently been injured, with injury resistant runners being intermediate of the two groups. It is also hypothesized that female runners will demonstrate significantly greater impact acceleration (Peak$_{accel}$ and Rate$_{accel}$) compared to males.

## Materials and methods

### Study design

This study was an early sub-study of a larger prospective longitudinal trial of recreational runners, examining the musculoskeletal, biomechanical and injury history risk factors of running-related injuries over an 12-month period (NCT03671395 www.clinicaltrials.gov). This study was approved by Dublin City University Research Ethics Committee, with written informed consent obtained from all participants prior to the study beginning (DCUREC/2017/186).

### Participants

Male and female recreational runners, aged between 18 and 65 years, who typically ran a minimum of 10km per week for the past 6 months [11], were recruited from local running events, running clubs, social media recruitment drives and radio advertising between January and August 2018. Participants were excluded if they were currently injured or had sustained an injury within the 3 months prior to testing [7], had a history of cardiovascular pathology, previous reconstructive joint surgery or joint replacement, or were pregnant. An online survey was given to eligible participants to gather information regarding their training history (weekly miles, quarterly annual miles, training speed and years running experience), and previous running injury history within the past two years. An RRI definition was adapted from a consensus statement, and was defined as "any running-related (training or competition) muscle, bone, tendon or ligament pain in the lower back/legs/knee/foot/ankle that caused a restriction or stoppage of running (distance, speed, duration or training) for at least 7 days or 3 consecutive

scheduled training sessions, or that required the runner to consult a physician or other health professional" [30, 31]. *An a-priori* (alpha probability = 0.05, with a power of 1- ß = 0.80, effect size $f$ = 0.25) statistical power analysis for a two-way ANOVA was performed using a G*Power program (G*Power 3.1.9.7) to determine the required sample size [32]. A total of 128 participants would be the minimum number of participants necessary. Three participant groups were constructed using the injury history data: recreational runners who were never injured (group 1) were matched by sex, quarterly annual mileage and typical training speed with runners who had acquired injury resistance (group 2; runners who have not been injured in the past 2 years), and runners who had been returned to running following a recent RRI (group 3; RRI 3–12 months prior to testing). Where more than one recently injured or acquired injury resistant runner could be matched to the never injured runner, the runner was chosen at random by flipping a coin, so as to eliminate bias from the matching selection. Runners who had been injured 1–2 years pre-testing were excluded from selection in order to ensure a clear demarcation between the "injury resistant" and "recently injured" running groups [12].

## Procedures

Participants signed an informed consent form on their initial visit to the laboratory. Prior to any physical testing, the primary researchers checked the survey responses for accuracy and completion, with all injury and training behaviour responses clarified with participants. Height (cm) (Leicester Height Measure, SECA, UK), body mass (kg) (SECA, UK), and limb dominance were recorded. Limb dominance was determined as the leg that the participant would choose to kick a football [33]. Inertial sensors (Shimmer3 IMU, Shimmer™, Ireland) containing accelerometers were used to capture (512Hz sampling rate) the magnitude (Peak$_{accel}$) and rate (Rate$_{accel}$) of impact acceleration of the tibia bilaterally, as well as for the sacrum. Two inertial measurement units were attached bilaterally 5 cm proximal to the medial malleolus using double-sided sticky tape, with the y-axis of the sensor aligned with the long axis of the shank [28]. They were then tightly secured using Hypafix adhesive tape which wrapped and adhered directly to the skin. The sacrum sensor was held in place within a custom-made elastic belt, with the longitudinal axis aligned to the vertical midline of the S2 spinous process [34]. The belt was attached to the skin over the sacrum using double-sided sticky tape, and this was secured further by tape and an elastic waistband on top. Securing the inertial sensors with double-sided sticky tape and wrapping has been found to be more representative of tibial accelerations when compared to less secure methods such as the manufacturer provided straps [35]. Running trials were conducted on a treadmill (Flow Fitness, Runner DTM3500i, The Netherlands) at a set speed of 9km/hr. The set speed of 9km/hr was chosen to allow for comparison of impact accelerations without the confounding factor of variations in speed affecting the participants' technique. This speed represented the average five-kilometre time of runners in the greater Dublin area, determined from the average speed reported on the Dublin Park Run database (www.parkrun.ie/events). During the testing session, once sensors had been attached and secured, participants completed a 5 minute warm up consisting of dynamic stretches for the hamstrings, quadriceps, hip flexors, hip extensors and calf muscle groups [36]. Participants then ran at 9km/hr for 6 minutes to ensure familiarisation to treadmill running [37]. Following the 6 minutes of familiarisation, the participants continued to run at 9km/hr for an additional 1 minute. This 1 minute period was chosen as the period for impact acceleration data extraction, and was standard for all participants. Participants were encouraged to continue running beyond this time, so that they would be blinded to the specific time period of data collection. Participants ceased running when they felt comfortable to do so, provided they had ran for a minimum of 7 minutes (6 minutes familiarisation + 1 minute data collection).

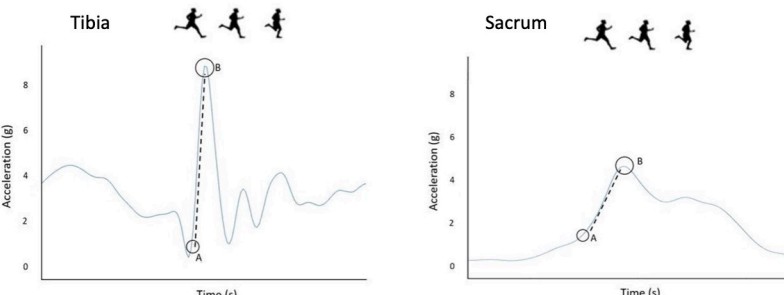

**Fig 1.** Trace of $Peak_{accel}$ and $Rate_{accel}$ for the shank (left) and sacrum (right). (A): initial contact detected; dotted line ----: $Rate_{accel}$, which was calculated as the slope of the peak (B).

## Data processing

Axial $Peak_{accel}$ and $Rate_{accel}$ of the shanks and sacrum were processed using a custom-built MATLAB script (Mathworks Inc., Natick, MA, USA). A fourth order, zero lag 60 Hz Butterworth filter was applied to the data, as documented in previous research [28] and dropped packets were filled using a cubic spline. To ensure functionally equivalent values were extracted from the shank and sacrum sensors, the time series data were time-aligned using the custom-built MATLAB script. $Peak_{accel}$ was taken as the maximal amplitude of the accelerometer's local maxima at initial contact and was expressed in units of standard gravity (g = 9.8 m/s$^2$). A series of pilot studies were conducted to identify initial contact utilizing a pressure sensitive switch in combination with inertial sensors, identifying robust patterns within the data. $Rate_{accel}$ was calculated as the slope of the $Peak_{accel}$ (Fig 1). Ten consecutive foot-strikes, taken immediately after the 6-minute familiarization, were processed on both dominant and non-dominant limbs.

## Statistical analysis

Descriptive statistics were used to summarize demographics, anthropometrics, and training data. A two-way between groups ANOVA (3 x 2) (injury status group x sex) was used to screen for significant differences in age, anthropometrics (height, weight and BMI), quarterly annual mileage and average running speed. Years running experience was captured nominally (i.e. 1–2 years; 3–5 years; 6–10 years, 11–15 years; 15 years +), and a Pearson Chi Square test was used to determine if significant differences in the number of years running experience existed between injury status groups. Boxplots were used to identify outliers that were 1.5 times the interquartile range above the upper quartile and below the lower quartile, with data outside these thresholds removed from the analysis [3]. To determine if there was a significant difference in impact acceleration between the dominant and non-dominant limbs, paired sample t-tests were employed. If no differences between limbs existed, dominant and non-dominant limbs would be pooled as one measure.

A two-way between groups ANOVA (3 x 2) (injury status group x sex) was conducted to examine differences in impact accelerations ($Peak_{accel}$ and $Rate_{accel}$) at the tibia and sacrum. Homogeneity of variance was assessed by Levene's test for equality of variances. The three injury status groups were: never injured runners (runners with no history of injury), runners with an acquired injury resistance (runners who have not been injured in the past 2 years), and recently injured runners (runners who had returned to running following an injury 3–12 months ago). Tukey HSD post-hoc tests were employed to identify differences between groups. The mean, standard deviation and effect size (partial eta squared) were reported using

the classification proposed by Cohen [38]; [trivial effect size = 0.00–0.19; small effect size 0.20–0.49; moderate effect size = 0.50–0.79, and large effect size = $\geq$ 0.80]. The alpha level for statistical significance was $p < .05$.

## Results

### Demographics

One hundred and forty seven (84 males, 63 females) recreational runners participating in a larger study (n = 310) were chosen in order to directly match participants across the three groups. A total of 49 recreational runners (28 male, 21 female) were identified as having never sustained an RRI. These 49 never injured runners were matched by sex, quarterly annual mileage, and typical training speed with 49 runners who had developed injury resistance, and with 49 runners who had recovered from a recent RRI 3–12 months before testing (Table 1). Participants ran on the treadmill for a mean time of 12 minutes and 32 seconds (± 5 minutes and 31 seconds). A breakdown of the RRIs sustained by the recently injured group can be viewed in Table 2. The knee was the most commonly injured region (23%), followed by the calf (20%) and foot (15%). Females had significantly lower weight, height, BMI and average training speeds than males ($p < .05$) (Table 1). No significant differences were found between the three injury groups for any of the demographic and training measures (age, weight, height, BMI, years running experience, quarterly annual mileage or average training speed) ($p > .05$) (Table 1). No significant differences were found for years running experience between the three injury status groups ($p = .78$).

### Impact acceleration

No significant differences were found between the dominant and non-dominant limbs for Peak$_{accel}$ or Rate$_{accel}$ of the tibia and sacrum ($p > 0.05$), and so the dominant and non-dominant limbs were pooled for subsequent analysis. The mean and standard deviation of impact acceleration results are presented in Table 3. No interaction effect was found between injury status and sex for any of the measures (tibia Peak$_{accel}$, tibia Rate$_{accel}$, sacrum Peak$_{accel}$ or sacrum Rate$_{accel}$) (Table 4). A significant main effect was found for injury status for sacrum Peak$_{accel}$ and sacrum Rate$_{accel}$ with trivial effect sizes (Table 4). Tukey post-hoc comparisons for sacrum Peak$_{accel}$ did not identify a significant difference between the three groups,

**Table 1. Participant demographics (mean ± standard deviation).**

| Demographics | Never Injured (n = 49) | | Injury Resistant (n = 49) | | Recently Injured (n = 49) | | Injury Status (P-value) | Sex (P-value) | Injury Status x Sex (P-value) |
|---|---|---|---|---|---|---|---|---|---|
| Sex | Male (n = 28) | Female (n = 21) | Male (n = 28) | Female (n = 21) | Male (n = 28) | Female (n = 21) | N/A | N/A | N/A |
| Age (years) | 43.6 ± 11.7 | 40.2 ± 8.2 | 43.6 ± 8.0 | 42.7 ± 9.2 | 43.0 ± 6.3 | 45.4 ± 6.4 | .435 | .627 | .244 |
| Weight (kg) | 81.8 ± 10.1$^\S$ | 59.9 ± 6.8$^\S$ | 82.1 ± 10.7$^\S$ | 61.5 ± 8.3$^\S$ | 80.6 ± 9.5$^\S$ | 61.5 ± 8.3$^\S$ | .877 | .000$^\S$ | .753 |
| Height (m) | 1.8 ± 0.1$^\S$ | 1.6 ± 0.1$^\S$ | 1.8 ± 0.1$^\S$ | 1.7 ± 0.1$^\S$ | 1.8 ± 0.1$^\S$ | 1.6 ± 0.1$^\S$ | .487 | .000$^\S$ | .395 |
| BMI (kg/m$^2$) | 26.0 ± 3.1$^\S$ | 22.3 ± 2.0$^\S$ | 25.7 ± 3.1$^\S$ | 22.6 ± 2.7$^\S$ | 25.2 ± 2.3$^\S$ | 23.5 ± 2.7$^\S$ | .933 | .000$^\S$ | .189 |
| Quarterly Annual Mileage (km) | 390.6 ± 246.7 | 354.8 ± 318.0 | 371.1 ± 254.2 | 368.6 ± 245.0 | 377.4 ± 227.2 | 342.3 ± 234.3 | .667 | .157 | .935 |
| Average Training Speed (km/hr) | 11.4 ± 2.1$^\S$ | 9.9 ± 2.7$^\S$ | 11.6 ± 1.7$^\S$ | 11.0 ± 1.5$^\S$ | 11.6 ± 1.6$^\S$ | 10.8 ± 1.5$^\S$ | .361 | .007$^\S$ | .478 |

N: number of participants; kg: kilogram; m: metre; kg/m2: kilogram per metre squared; km: kilometre; km/hr: kilometres per hour; P-value: significance level of $p < .05$
$^\S$: significant difference between males and females ($p < .05$); N/A: not applicable.

**Table 2. Breakdown of injury locations in the recently injured group.**

|  | Male: n (%) | Female: n (%) | All: n (%) |
|---|---|---|---|
| Knee | 8 (19.0%) | 7 (30.4%) | 15 (23.1%) |
| Calf/Achilles | 9 (21.4%) | 4 (17.4%) | 13 (20.0%) |
| Foot | 7 (16.7%) | 3 (13.0%) | 10 (15.4%) |
| Lower Back & SIJ | 8 (19.0%) | 1 (4.4%) | 9 (13.9%) |
| Posterior Thigh | 2 (4.8%) | 3 (13.0%) | 5 (7.7%) |
| Hip & Buttock | 2 (4.8%) | 2 (8.7%) | 4 (6.2%) |
| Shin | 3 (7.1%) | 1 (4.4%) | 4 (6.2%) |
| Ankle | 1 (2.4%) | 2 (8.7%) | 3 (4.6%) |
| Groin | 2 (4.8%) | 0 (0.0%) | 2 (3.1%) |
| *Total* | 42 (100%)^ | 23 (100%)^ | 65 (100%)^ |

N: number of injuries; ^: 65 injuries between 49 runners– 36 runners sustained 1 RRI, 11 runners sustained 2 RRIs, 1 runner sustained 3 RRIs and 1 runner sustained 4 RRIs; SIJ: sacroiliac joint; All: males and females combined

^Note: percentages may not add up to 100% as values were rounded up to 1 decimal place.

however, the greater mean impact acceleration observed between the recently injured group compared to the acquired injury resistance group approached statistical significance (p = .061). Tukey post-hoc comparisons for sacrum $Rate_{accel}$ indicated that the mean impact acceleration for the recently injured group was significantly greater than both the never injured group and the acquired injury resistance group. A significant main effect for sex was found for tibia $Peak_{accel}$, tibia $Rate_{accel}$, and sacrum $Rate_{accel}$ with trivial effect sizes, with sacrum $Peak_{accel}$ approaching significance (p = .07) (Table 4). Females demonstrated significantly greater $Peak_{accel}$ and $Rate_{accel}$ at the tibia and significantly greater sacrum $Rate_{accel}$ than their male counterparts.

## Discussion

This study hypothesized that runners who have never been injured would demonstrate significantly lower impact acceleration ($Peak_{accel}$ and $Rate_{accel}$) compared to runners who had recently been injured, with injury resistant runners being intermediate of the two groups. It

**Table 3. Mean and standard deviation of $Peak_{accel}$ and $Rate_{accel}$ for the tibia and sacrum.**

|  | Never Injured | | | Injury Resistant | | | Recently Injured | | |
|---|---|---|---|---|---|---|---|---|---|
| Impact Acceleration | All | Males | Females | All | Males | Females | All | Males | Females |
| Tibia $Peak_{accel}$ (g) | 5.84 ± 1.63 | 5.54 ± 1.16[§] | 6.22 ± 2.06[§] | 6.07 ± 1.47 | 5.53 ± 1.07[§] | 6.84 ± 1.64[§] | 5.92 ± 1.61 | 5.47 ± 1.10[§] | 6.48 ± 1.97[§] |
| *Range* | 3.8–10.3 | 3.8–8.1 | 3.8–10.3 | 3.8–10.2 | 3.8–8.3 | 3.8–10.2 | 3.6–10.2 | 3.8–8.3 | 3.6–10.2 |
| Tibia $Rate_{accel}$ (g/s) | 409.2 ± 179.9 | 382.1 ± 123.7[§] | 445.4 ± 234.4[§] | 470.3 ± 204.3 | 398.7 ± 239.5[§] | 571.7 ± 239.5[§] | 439.9 ± 195.0 | 397.4 ± 148.8[§] | 494.6 ± 234.5[§] |
| *Range* | 153.3–936.5 | 170.0–678.2 | 153.3–936.5 | 187.0–886.5 | 188.3–656.6 | 187.0–886.5 | 134.4–1118.5 | 134.4–710.9 | 138.5–1118.5 |
| Sacrum $Peak_{accel}$ (g) | 5.53 ± 1.60 | 5.29 ± 1.58 | 5.86 ± 1.59 | 5.34 ± 2.02 | 5.26 ± 1.83 | 5.45 ± 2.29 | 6.18 ± 1.76 | 5.82 ± 1.65 | 6.71 ± 1.83 |
| *Range* | 0.8–9.0 | 3.1–8.9 | 0.8–9.0 | 2.0–9.4 | 2.0–9.4 | 2.1–8.7 | 2.8–10.0 | 2.8–8.1 | 3.4–10.0 |
| Sacrum $Rate_{accel}$ (g/s) | 253.5 ± 140.5* | 229.5 ± 131.2[§] | 284.2 ± 149.2[§] | 239.4 ± 139.1* | 220.1 ± 128.4[§] | 265.1 ± 151.6[§] | 326.4 ± 170.9* | 253.9 ± 111.0[§] | 428.0 ± 190.1[§] |
| *Range* | 34.0–739.5 | 74.1–596.0 | 34.0–739.5 | 37.1–587.3 | 37.1–587.3 | 72.9–494.1 | 105.0–660.3 | 105.0–482.1 | 118.8–660.3 |

$Peak_{accel}$: magnitude of acceleration; $Rate_{accel}$: rate of acceleration; g: g force; g/s: g force per second; All: Inclusive of both males and females

*: significant difference between injury status groups as identified in post-hoc analysis at p < .05

[§]: significant difference between males and females.

**Table 4. Results of the two-way ANOVA investigating the differences between injury status and sex for impact acceleration.**

| | Injury Status | | Sex | | Injury Status x Sex interaction | |
|---|---|---|---|---|---|---|
| Impact Acceleration | P value | Effect Size | P value | Effect Size | P value | Effect Size |
| Tibia Peak$_{accel}$ | .611- | *.007* | .000* | .103 (Trivial) | .588 | *.008* |
| Tibia Rate$_{accel}$ | .190- | *.024* | .001* | .084 (Trivial) | .361 | *.015* |
| Sacrum Peak$_{accel}$ | .043* | *.045 (Trivial)* | .072- | .023 | .643 | *.006* |
| Sacrum Rate$_{accel}$ | .002* | *.086 (Trivial)* | .000* | .095 (Trivial) | .053 | *.041* |

Peak$_{accel}$: magnitude of acceleration; Rate$_{accel}$: rate of acceleration

*: significant p value at $p < .05$.

was also hypothesized that female runners would demonstrate significantly greater impact acceleration (Peak$_{accel}$ and Rate$_{accel}$) compared to males. The findings partly support the primary hypothesis, with results indicating that runners who have recently been injured demonstrated significantly greater Rate$_{accel}$ at the sacrum than runners who had never been injured, with runners who had acquired injury resistance being intermediate of the two groups. Although there was a significant main effect for injury status on sacrum Peak$_{accel}$, the post-hoc analysis did not reach significance ($p = .06$). There was no significant difference in tibia Peak$_{accel}$ or Rate$_{accel}$ between the three injury groups. Thus, it appears that measures at the sacrum are more sensitive to injury status than the tibia. A previous study by Schütte *et al.*, [23] observed a similar level of difference (10.0%) in sacrum Peak$_{accel}$ to our study (10.5%) between recently injured and uninjured runners, but no previous research has been conducted with respect to sacrum Rate$_{accel}$, and so comparison of these findings cannot be drawn.

Based upon the findings of our study, it appears that the never injured and acquired injury resistance runners use a technique that produces lower impact acceleration rates at the sacrum. Given that the never injured group demonstrated the lowest Rate$_{accel}$ at the sacrum, it seems that this low loading rate is protective against the likelihood of RRIs. For runners who have acquired injury resistance, this group may have adapted a strategy to reduce their Rate$_{accel}$ when returning to running after injury, ultimately aiming to alleviate excessive load on weakened or damaged structures, and to reduce their likelihood of sustaining subsequent RRIs. Perhaps the presence of high Rate$_{accel}$ in the recently injured group demonstrates a failure to adapt such a strategy and may indicate why this group has been injured most recently from the time of testing. Evidence of this has been demonstrated previously, where currently injured runners have demonstrated significantly greater vGRF loading rates compared to injury-free runners [39]. However, research to date has not captured impact loading across a continuous injury timeline (pre-injury, presence of injury and post-injury), and so this is only speculation of the potential injurious mechanisms and recovery strategies at play. It is important to consider that the recently injured group will inevitably develop into either a re-injury group or an injury-resistant group, and so future studies should track these individuals to see if there are ways to identify those who become re-injured and those who don't. A recent study has found hinderance from a previous injury to be highly associated with the occurrence of a subsequent RRI [40], suggesting that runners may have returned to running without addressing the potential biomechanical factors that might have contributed to their initial injury. Considering the sensitivity of sacral impact accelerometers in distinguishing between injury groups in this study, there are prospects for runners to be more objectively guided in their return to running following RRIs.

In contrast to the findings at the sacrum, no significant main effects for injury status on Peak$_{accel}$ or Rate$_{accel}$ were evident at the tibia, which partially rejects the primary hypothesis.

Although there are no previous studies that have investigated tibial $Rate_{accel}$ between injured and uninjured runners, there are mixed findings in the literature regarding differences between tibial $Peak_{accel}$ in injured and uninjured runners. The results of this study are in agreement with some studies that found no significant difference in tibia $Peak_{accel}$ between recently injured and uninjured runners [23–25]. Conversely, our findings disagree with the results of Milner *et al.*, [3] and Ferber *et al.*, [16], who both found $Peak_{accel}$ at the tibia to be significantly greater in female runners with a history of lower limb stress fractures compared to uninjured runners. This contrast in findings may be due to two reasons. Firstly, the primary aim of our study was to compare impact acceleration in runners with a history of any overuse RRIs rather than focusing directly on specific RRIs such as lower limb stress fractures. Perhaps measures of $Peak_{accel}$ at the tibia are more sensitive in differentiating between runners who have a history of local injury to the tibia itself [3], rather than differentiating between general overuse RRIs. Secondly, the secondary aim of this study was to determine the interaction effect of sex on injury status with respect to impact acceleration, necessitating the inclusion of male runners in our analysis.

A secondary hypothesis of this study was that female runners would demonstrate significantly greater impact acceleration ($Peak_{accel}$ and $Rate_{accel}$) at the tibia and sacrum compared to males. While there was no interaction effect between sex and injury status, sex was a main effect with significantly larger tibial $Peak_{accel}$ (11–19%), tibial $Rate_{accel}$ (14–30%) and sacrum $Rate_{accel}$ (17–41%) evident for females compared to males. In addition, differences between females and males for sacrum $Peak_{accel}$ (4–13%) approached significance (p = .07). Little research has been devoted to investigating the differences in impact acceleration between sexes during running, but the results of this study are similar to some previous findings where females have demonstrated greater $Peak_{accel}$ at the tibia [41] and sacrum [42] compared to males. As stated previously, $Rate_{accel}$ has not been a focus of research to date, but differences in vGRF loading rates were similarly greater in females compared to males in previous studies [3, 43, 44]. Differences in running kinematics (e.g. greater hip adduction) [45], muscle contractions (e.g. delayed gluteus medius activation) [46] and lower body alignment (e.g. greater tibia varum) [47] in females compared to males have been proposed as potential reasons for the higher impact accelerations in females [42, 48]. The factors mentioned above have been shown to relate to specific RRIs such as patellofemoral pain syndrome [49], iliotibial band friction syndrome [50] and stress fractures [51], potentially leading to an increased predisposition of specific injuries for female runners [45]. Given that the present study examined retrospective injuries, further prospective studies are required to investigate the impact acceleration differences between males and females, how this impact accelerations are affected by biomechanics, and if these factors relate to prospective injury occurrence.

## Limitations

There are some limitations to this study, one of which is the retrospective nature of the analysis. Although this study provides a unique insight into novel injury groups (never injured and injury resistant runners), future research should examine the relations between segmental impact loading and RRI prospectively. Secondly, the injury history for this study was self-reported, and therefore may be subject to recall bias or inaccuracies. In efforts to minimize this, the side of injury and exact pathology of each RRI was not collated, and RRIs were grouped by general location.

## Conclusion

This study found $Rate_{accel}$ at the sacrum to be significantly greater in recently injured runners compared to runners with acquired injury resistance and never injured runners. These

findings suggest that Rate$_{accel}$ at the sacrum is an appropriate objective measure to distinguish recently injured runners, potentially informing rehabilitation goals for runners with higher rates of acceleration at the sacrum when returning to running following RRIs. Examples of gait re-training for impact acceleration attenuation have been observed through the literature [52–54], and have proven to be effective in reducing impact loading in both injured and uninjured populations.

This study also found females to demonstrate significantly greater Peak$_{accel}$ and Rate$_{accel}$ at the tibia, and Rate$_{accel}$ at the sacrum than their male counterparts. As repetitive loading is thought to be an influential factor in RRI development, females with greater impact acceleration, or poor impact attenuation capacity may therefore be at increased susceptibility to overuse RRIs (e.g. stress fractures). This may indicate a clinical use for impact accelerometers in gait re-education for impact attenuation and potential injury prevention in female runners.

## Acknowledgments

The authors thank the participants for their time in partaking in this study.

## Author Contributions

**Conceptualization:** Aoife Burke, Sarah Dillon, Siobhán O'Connor, Enda F. Whyte, Shane Gore, Kieran A. Moran.

**Data curation:** Aoife Burke, Sarah Dillon.

**Formal analysis:** Aoife Burke, Shane Gore.

**Funding acquisition:** Kieran A. Moran.

**Investigation:** Aoife Burke, Sarah Dillon.

**Methodology:** Aoife Burke, Sarah Dillon, Siobhán O'Connor, Enda F. Whyte, Shane Gore, Kieran A. Moran.

**Project administration:** Aoife Burke, Sarah Dillon.

**Resources:** Aoife Burke.

**Visualization:** Aoife Burke.

**Writing – original draft:** Aoife Burke.

**Writing – review & editing:** Aoife Burke, Sarah Dillon, Siobhán O'Connor, Enda F. Whyte, Shane Gore, Kieran A. Moran.

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
