## [Decision Letter · Decision Letter 0]

18 Apr 2022

PONE-D-22-06813Comparison of impact accelerations between injury-resistant and recently injured recreational runnersPLOS ONE

Dear Dr. Burke,

Thank you for submitting your manuscript to PLOS ONE. After careful consideration, we feel that it has merit but does not fully meet PLOS ONE’s publication criteria as it currently stands. Therefore, we invite you to submit a revised version of the manuscript that addresses the points raised during the review process.

We look forward to receiving your revised manuscript.

Kind regards,

Shazlin Shaharudin

Academic Editor

PLOS ONE

Journal Requirements:

"This study has emanated from research supported by Science Foundation Ireland (SFI) under grant number SFI/12/RC/2289_P2, cofounded by the European Regional Development Fund. The authors thank the participants for their time in partaking in this study."

"This study has emanated from research supported by Science Foundation Ireland (SFI) under grant number SFI/12/RC/2289_P2, cofounded by the European Regional Development Fund. The authors thank the participants for their time in partaking in this study."

"The authors declare that they have no competing interests. The results of the study are presented clearly, honestly, and without fabrication, falsification, or inappropriate data manipulation."

Reviewers' comments:

Reviewer's Responses to Questions

**Comments to the Author**

1. Is the manuscript technically sound, and do the data support the conclusions?

Reviewer #1: Yes

Reviewer #2: Yes

Reviewer #3: Yes

2. Has the statistical analysis been performed appropriately and rigorously? 

Reviewer #1: Yes

Reviewer #2: I Don't Know

Reviewer #3: Yes

3. Have the authors made all data underlying the findings in their manuscript fully available?

Reviewer #1: Yes

Reviewer #2: Yes

Reviewer #3: Yes

4. Is the manuscript presented in an intelligible fashion and written in standard English?

Reviewer #1: Yes

Reviewer #2: Yes

Reviewer #3: Yes

5. Review Comments to the Author

Reviewer #1: The paper is generally well written and structured. The objective of this study is to determine the Peakaccel and Peakrate of the tibia and sacrum between three different groups and author has shown enough data to support the hypothesis. However, some of my comments are:

1. Participants were recruited from online survey, and how does one ensure participants were well versed with the questions from the survey especially those with medical terms (for injury) to reduce biased and error?

2. The paper compares the impact of accelerations in runners with history of any overuse RRI and not specific to injury locations directly, does this effect the overall data other than mentioned in L291 – 298?

Reviewer #2: I have included my comments as an attachment with this submission as a separate file. Please check all the comments for further details. Thank you for allowing me to review the manuscript.

Reviewer #3: No further comments, well done. I have just completed reading your article, everything was good and perfect. Appreciate a research article in your own good way. A token of appreciation can be how you feel can be expressed the best. It was an exceptionally well-written article and created many interesting suggestions on the subject.

6. PLOS authors have the option to publish the peer review history of their article (what does this mean?). If published, this will include your full peer review and any attached files.

Reviewer #1: No

Reviewer #2: No

Reviewer #3: **Yes: **FARA LIANA BINTI ZAINUDDIN

---

## [Author Response · Author response to Decision Letter 0]

20 Jun 2022

Many thanks for providing such valuable feedback and review on our recently submitted manuscript to Plos One. We have tried to amend the manuscript, and have addressed all comments and feedback within the table in the following pages. We hope our responses are satisfactory, but should you need further clarification on anything, please do not hesitate to contact me.

---

## [Decision Letter · Decision Letter 1]

26 Jul 2022

PONE-D-22-06813R1Comparison of impact accelerations between injury-resistant and recently injured recreational runnersPLOS ONE

Dear Dr. Burke,

Thank you for submitting your manuscript to PLOS ONE. After careful consideration, we feel that it has merit but does not fully meet PLOS ONE’s publication criteria as it currently stands. Therefore, we invite you to submit a revised version of the manuscript that addresses the points raised during the review process.

ACADEMIC EDITOR:  Please answer the additional questions that reviewer 2 has asked.

We look forward to receiving your revised manuscript.

Kind regards,

Javier Abián-Vicén, Ph.D.

Academic Editor

PLOS ONE

Journal Requirements:

Reviewers' comments:

Reviewer's Responses to Questions

**Comments to the Author**

1. If the authors have adequately addressed your comments raised in a previous round of review and you feel that this manuscript is now acceptable for publication, you may indicate that here to bypass the “Comments to the Author” section, enter your conflict of interest statement in the “Confidential to Editor” section, and submit your "Accept" recommendation.

Reviewer #1: All comments have been addressed

Reviewer #2: (No Response)

2. Is the manuscript technically sound, and do the data support the conclusions?

Reviewer #1: Yes

Reviewer #2: Yes

3. Has the statistical analysis been performed appropriately and rigorously? 

Reviewer #1: Yes

Reviewer #2: I Don't Know

4. Have the authors made all data underlying the findings in their manuscript fully available?

Reviewer #1: Yes

Reviewer #2: (No Response)

5. Is the manuscript presented in an intelligible fashion and written in standard English?

Reviewer #1: Yes

Reviewer #2: Yes

6. Review Comments to the Author

Reviewer #1: Author able to addressed all comments from previous review accordingly. Manuscripts has been amended splendidly.

Reviewer #2: Thank you for addressing the reviewer's comments. The additional comments have been attached as a separate file.

7. PLOS authors have the option to publish the peer review history of their article (what does this mean?). If published, this will include your full peer review and any attached files.

Reviewer #1: No

Reviewer #2: No

---

## [Author Response · Author response to Decision Letter 1]

9 Aug 2022

Many thanks for your recent response. Amendments have been made to the manuscript as per the comments of Reviewer #2.

Should you have any further changes or comments, please do not hesitate to contact me.

---

## [Editor Report · Decision Letter 2]

12 Aug 2022

Comparison of impact accelerations between injury-resistant and recently injured recreational runners

PONE-D-22-06813R2

Dear Dr. Burke,

We’re pleased to inform you that your manuscript has been judged scientifically suitable for publication and will be formally accepted for publication once it meets all outstanding technical requirements.

Kind regards,

Javier Abián-Vicén, Ph.D.

Academic Editor

PLOS ONE

---

## [Editor Report · Acceptance letter]

1 Sep 2022

PONE-D-22-06813R2 

Comparison of impact accelerations between injury-resistant and recently injured recreational runners 

Dear Dr. Burke:

I'm pleased to inform you that your manuscript has been deemed suitable for publication in PLOS ONE. Congratulations! Your manuscript is now with our production department. 

Kind regards, 

on behalf of

Dr. Javier Abián-Vicén 

Academic Editor

PLOS ONE